# Impact of In Vitro Degradation on the Properties of Samples Produced by Additive Production from PLA/PHB-Based Material and Ceramics

**DOI:** 10.3390/polym14245441

**Published:** 2022-12-12

**Authors:** Alena Findrik Balogová, Marianna Trebuňová, Darina Bačenková, Miroslav Kohan, Radovan Hudák, Teodor Tóth, Marek Schnitzer, Jozef Živčák

**Affiliations:** Department of Biomedical Engineering and Measurement, Faculty of Mechanical Engineering, Technical University of Kosice, Letná 1/9, 042 00 Košice, Slovakia

**Keywords:** biodegradation, in vitro, polymer, ceramics

## Abstract

The present study deals with preparing a polymer-based material with incorporated ceramics and monitoring changes in properties after in vitro natural degradation. The developed material is a mixture of polymers of polylactic acid and polyhydroxybutyrate in a ratio of 85:15. Ceramic was incorporated into the prepared material, namely 10% hydroxyapatite and 10% tricalcium phosphate of the total volume. The material was processed into a filament form, and types of solid and porous samples were prepared using additive technology. These samples were immersed in three different solutions: physiological solution, phosphate-buffered saline, and Hanks’ solution. Under constant laboratory conditions, changes in solution pH, material absorption, weight loss, changes in mechanical properties, and surface morphology were monitored for 170 days. The average value of the absorption of the solid sample was 7.07%, and the absorption of the porous samples was recorded at 8.33%, which means a difference of 1.26%. The least change in pH from the reference value of 7.4 was noted with the phosphate-buffered saline solution. Computed tomography was used to determine the cross-section of the samples. The obtained data were used to calculate the mechanical properties of materials after degradation. The elasticity modulus for both the full and porous samples degraded in Hanks’ solution (524.53 ± 13.4 MPa) has the smallest deviation from the non-degraded reference sample (536.21 ± 22.69 MPa).

## 1. Introduction

The Miller-Keane Encyclopaedia states that a biomaterial can be defined as any substance, whether of natural or synthetic origin, that can be used as a system or as part of a system to treat and replace the function of a tissue, organ, or body. Of course, it is important to consider that this is a material that is in contact with living tissue [1].

In tissue engineering and regenerative medicine, materials suitable for implantation are increasingly being studied, with the goal that there would be no need for secondary intervention to remove them. Thus, work is being done on materials that, after implantation, degrade while new tissue is created. Therefore, it is important to know the degradation properties of such material in an environment similar to the human body.

According to a study by Victor et al. [2], currently polymer–ceramic composites are considered third-generation orthopedic biomaterials because they have the ability to match the natural properties of bones compared to bone substitutes of the first and second generation. The addition of selected composites to polymers, which will thus create materials with mechanical properties comparable to those of bone while also supporting the bone formation, is a major challenge today, since the addition of composites to polymers may alter the degradation properties of final composite materials [3,4,5] and may change the pH of the surrounding environment as well [6]. It is important to establish the correct ratio of the individual components of the selected studied material and to carry out as many measurements as possible related to the degradation of the generated material, evaluating the influence of the material on the pH of the environment as well as determining its mechanical properties, cytotoxicity, and biocompatibility.

The bone replacement implant should be partially resorbable, but at the same time, the ceramic component of the material should provide support for osteoinductivity, so it is essential to investigate the degradation behavior of biodegradable materials. The rate of degradation is a critical factor influencing the healing of bone fractures. Several studies have focused on in vitro and in vivo degradation of poly L-lactic acid (PLLA) based composites. However, there are conflicting results on the effect of the bioceramic filler on the resorption rate. Some researchers have reported that the addition of a ceramic component slowed down the degradation of the composite. For example, Bleach et al. [3] found that PLLA absorbed more water and showed more significant mass loss than samples containing hydroxyapatite (HA) or calcium phosphate (TCP) after immersion in simulated body fluid (SBF) for 12 weeks. Niemelä et al. [4] reported that the degradation of the β-TCP/PLA composite was slower than that of polylactide (PLA). Araújo et al. [7] observed that incorporating a clay mineral into the PLA matrix increased the thermal stability of the polymer.

In contrast, other authors have noted an increase in the rate of degradation in the presence of HA, TCP, or other ceramic components, attributed to the particle/matrix interface and the hydrophilicity of the ceramic. For example, Delabarde et al. [8] and Jiang et al. [9] stated that the incorporation of HA into HA/PLA (or HA/polylactic-co-glycolic acid (PLGA)) composites could accelerate degradation at the matrix/particle interface. The addition of β-TCP11 and soluble calcium phosphate glass (CaP) 12 has also accelerated PLA degradation.

The influence of the pH of the environment on the material and vice versa can affect the rate of degradation of the material. In a study by Lebo et al. [6], they monitored PLA material in phosphate-buffered saline (PBS) at 37 °C, evaluating PLA degradation depending on the pH of the environment. As a result, the degradation of PLA occurred faster in an environment with a higher pH (pH = 8). No degradation was observed at pH = 3. Thus, the pH value has a considerable impact on the degradation speed.

The degradation of polymer–ceramic composites considered as orthopaedic biomaterials can significantly affect three-dimensional cell growth and angiogenesis. These are two critical factors involved in bone regeneration [10]. In addition, the rate of degradation of polymer–ceramic composites could adversely affect the mechanical properties of the reconstructed bone. This could lead to poor bone regeneration and failure of integrity, especially in cases of nasal bone tissue applications such as spinal fusion implants [11].

The aim of this study is to evaluate the properties of a PLA/PHB material mixture with ceramics after in vitro degradation of the 3D printed sample under laboratory conditions. Two types of samples (porous and solid) were prepared from the material using 3D printing technology. The produced samples were subjected to analyses, where we determined the absorbency of the solid sample with the solution, and the degradation of the sample when exposed to a specific pH environment was evaluated. Computed tomography (CT) was used to determine the cross-section of the samples, and the obtained results were used to calculate the mechanical properties of the materials determined after degradation. The material was placed in three solutions for 170 days under constant laboratory conditions with a pH of 7.4 and a temperature of 37 °C.

## 2. Materials and Methods

The material used was developed by the Department of Biomedical Engineering and Measurement, Technical University of Košice. All tests such as in vitro degradation and biological testing were performed on materials consisting of a mixture of polymers of polylactic acid and polyhydroxybutyrate in a weight ratio of 85:15. In addition, ceramics were incorporated into the prepared amount, namely 10% hydroxyapatite and 10% tricalcium phosphate of the total volume. Subsequently, the prepared material was subjected to biodegradation under conditions that simulated the human body.

### 2.1. Filament Production

As fused deposition modeling (FDM) base additive technology was used to produce the samples, the material in the form of pellets was processed into a filament using COMPOSER 450 (3devo, Utrecht, Netherlands). In this case, the material passed through 4 extruder temperature zones, the temperatures of which were: zone 1: 175 °C, zone 2: 180 °C, zone 3: 175 °C, and zone 4: 155 °C. As a result, a filament with a diameter of 1.75 ± 0.05 mm was produced.

The detailed parameters of each component of the designed composite material are in Table 1 and Table 2.

### 2.2. Preparation and 3D Printing of Samples for In Vitro Degradation

The samples were designed in the Magics Materialize (Belgium) program in a cylindrical shape with a diameter of 5 mm and a height of 2 mm. The dimensions of the samples were defined based on several analyzed studies [12].

TRILAB DeltiQ (Hradec Kralove, Czech Republic) was used to print the samples. This printer is characterized by the so-called Delta kinematics that ensures precise and fast movement of the printer head throughout the volume. The basic printing parameters were set to a print head temperature of 220 °C, a platform temperature of 60 °C, nozzle diameter of 0.4 mm, and a print speed of 20 mm/s. The 3D printer created 180 samples, of which 90 were solid and 90 were porous. Subsequently, all samples were weighed sequentially on a MA 50/1.X2.A scale from Radwag (Radom, Poland).

The thickness of one layer during printing was given as 0.3 mm. The infill was 100% for solid and 50% for porous samples. In both types of specimens, the individual layers were turned 90° relative to the lower layer.

The printing parameters for this new material, which had not yet been processed by the 3D printing method, were determined empirically, and the determined values are given in Table 1.

### 2.3. Ensuring Constant Degradation Conditions

The printed samples were divided according to the structure (solid, porous) into the appropriate biodegradation media: physiological, PBS, or Hanks’ solution. A group of 30 samples were placed in each solution. Samples with the appropriate solutions in a volume of 40 mL were placed in beakers and then placed on the platform of an Orbital Shaker PSU-10i (BioSan, Riga, Latvia) with a stirring speed of 150 revolutions per minute (RPM), thus ensuring simulated fluid flow throughout the biodegradation period [5].

To provide additional simulated conditions, the pH of the solutions was set at 7.4. The temperature in the incubator CelCulture^®^ CO_2_ Incubator CCL-170B-8 (Esco Micro Pte. Ltd., Singapore), where the mixing device was placed with the examined samples in the beakers and had a constant temperature of 37 °C. The temperature was controlled by a device with a temperature fluctuation of 0.2 °C. During the experiment, the beakers were closed and there was only minimal evaporation of the solution. During the whole experiment, there was no need to replenish the solution. When weighing the samples, the pH of the solution was measured continuously, and in the event of a change in the pH levels, the pH of the solution was corrected.

The experiment lasted 170 days. The solutions were the same throughout the biodegradation period; they were not exchanged once. For the first 75 days, the solutions and samples were unmeasured. Then, every 14 days, the samples were regularly weighed and the pH was adjusted in all solutions using a Metter Toledo pH meter (Metter Toledo, Bratislava, Slovak Republic). From day 142, 15 samples were taken from each group and were used for biological testing and SEM microscopy.

### 2.4. Analysis of Changes in Sample Weight

#### 2.4.1. Absorption

The specimens were regularly taken out of the solutions during the experiment at 14-day intervals. Then, the fluid on their surface was removed using filtration paper, and the specimens were weighed. The absorption percentage of the specimens was calculated using the following formula [13,14]:Sw = ((Wwer − Wdry))/W(dry) × 100 [%](1)

The absorption of the solution by the specimens and by the granulate is characterized as the swelling percentage Sw, where Wwet is the weight of a specimen on a particular day during the biodegradation. On the other hand, Wdry is the weight of a specimen prior to biodegradation, i.e., the weight of a specimen immediately after 3D printing before it is first immersed in a solution.

#### 2.4.2. Weight Loss

After 170 days of the experiment, five samples were selected from each solution. These were washed in distilled water to remove degradation products or other impurity traces. Next, drying took place in a Binder incubator (Otto Bock Healthcare, Duderstadt, Germany) at 60 °C for 24 h. The temperature and drying time were selected from an experiment by El-Kady et al. [15]. After this step, the samples were weighed. The percentage weight loss due to biodegradation was calculated according to the formula [15]:wl = ((winitial − wdry)/winitial) × 100 [%](2)
where wl is the weight loss, winitial represents the median of the initial weights before biodegradation, and wdry is, in our case, the median of the dried samples.

### 2.5. Testing of Mechanical Properties of Samples

Mechanical tests in single-axis static pressure were carried out on the Hegewald & Peschke Inspekt Blue with a sensor range of 5 kN. The squeezing speed was 2 mm/min. The end of the test was set to a press of 1.5 mm or reaching a force of 4950 N. All samples measured were dried before mechanical testing. The results of mechanical testing of samples in static pressure are represented by the dependence of force (N) on the position of the crosshead (mm).

## 3. Results

Evaluations of all samples were performed under constant conditions.

### 3.1. Absorbation of Solutions by Materials

Samples of the appropriate material were weighed on an analytical balance after the 3D printing process. The measured values were recorded in tables. This process was repeated at two-week intervals. The attached graphs make it possible to observe the changes in weight caused by biodegradation.

For the calculation of each series, the degree of central tendency was determined; the so-called median. At the same time, it was necessary to consider the fact that the balance has measurement uncertainty. In our case, the uncertainty value was set at 0.001%, as 1.0000 g ± 0.01 mg represents 0.001%.

When looking at the average percentage of the weight gain of the solid and porous samples compared to their original weight before biodegradation (Table 3), changes are observed in both the structure of the samples and the used degradation media.

In the second type, the increase is higher in the case of the porous samples. Again, the reason is the structure. Since a lower infill was used for the porous samples, some of the solutions were absorbed into the internal structure earlier than for the solid samples.

The absorption of the solutions by the sample is directly proportional to the increasing weight of the samples.

Looking at the comparison within the solutions, we notice that the PBS solution shows a higher difference between the solid and porous samples than the others. Furthermore, it was confirmed by statistical analysis that the type of biodegradation medium and the shape of the sample itself affect the change in weight of the sample. The graph (Figure 1) shows the absorption trajectory of solid and porous samples in individual solutions. It shows that the absorption of the porous samples was significantly higher than the absorption of the solid samples. This fact applies to all biodegradation solutions. In terms of percentage evaluation, the average value of the absorption of the solid sample was 7.07% and the absorption of the porous samples was recorded at the level of 8.33%, which means a difference of 1.26%. Of all the average absorbances of the samples, the highest and lowest values were captured in PBS solution, namely 10.31% for porous samples and 4.88% for solid samples.

### 3.2. Evaluation of Weight Loss after Degradation

Table 4 shows that the weight loss due to biodegradation is only manifested in PBS solution. A more significant percentage difference occurred in the porous samples, as they degrade to a greater extent than the solid ones. This fact is unanticipated since the remnants of the coating were visible in all solutions, which indicates the effect of the ongoing degradation process.

### 3.3. Influence of Material on pH of Solution

As mentioned in the experiment, the pH values of all prepared solutions were measured and adjusted every 14 days to a value of 7.4. This measurement lasted 198 days. The measured values showed changes in the type of solution and the structure of the samples (porous vs. solid), which were degraded in these solutions. If we look at changes in the pH values of the exact solutions in which different samples degraded, we find that higher pH values were measured in solutions in which porous samples were present. In general, however, pH values were more likely to be less than 7.4. More significant fluctuations were observed in solid saline samples, where the values mainly indicated an acidic environment (Figure 2).

The PBS solution showed the most stable pH levels during biodegradation. In the attached graphs, a red line indicates a value of 7.4, which was determined as a constant for a given biodegradation process.

### 3.4. CT Surface Analysis

CT analysis was performed using a Metrotom 1500 from Carl Zeiss (Oberkochen, Germany) after the end of the experiment. Scanning was performed at a resolution of 24.3 µm. From each material and type of sample (full, porous), 1 pc was scanned and 1 non-degraded sample was taken as a reference. Figure 3 shows the location of the samples when capturing and surface reconstruction of all samples.

Using an industrial CT device, we obtained information on individual samples’ volume, surface, and radius (Table 5).

The non-degraded solid and porous samples’ radii clearly show that their values are lowered by 0.1 mm and 0.06 mm, respectively, from the sample radius reference value (i.e., 2.5 mm). The deviation occurred during printing, as the Trilab 3D printer used to print the samples has a resolution of 0.1 mm.

This analysis also confirmed that the samples absorbed some of the solutions volume in which they were present; as the samples volume, area, and diameter increased in the degraded samples.

The analysis showed that the samples were deformed in the degraded samples. Although they were horizontal along the edge, there was a more pronounced drop outside the center of the sample (Figure 4). The lower green line symbolizes the upper edge of the interlayer. The second line is at the level of the top surface of the sample, and the line indicates the drop of the layer at the center of the sample. The overflow in the intermediate layer is about 0.13 mm at the selected place. This phenomenon can also be observed on the top layer, by up to 0.23 mm. The material drop thus increases towards the higher layers.

In comparison to the non-degraded sample, the mentioned material overflow occurred due to degradation.

The diameter of the inner layer of the degraded sample ranged from 0.36 to 0.46 mm (Figure 5). Prior to measurement, the sample was aligned on the cylinder’s axis from the outer surface of the sample.

The diameter of the outer layer in the selected section, i.e., the section through the interlayer, was from 0.46 to 0.70 mm (see Figure 6). However, from above, the evaluation of the infill diameter was not appropriate, as it was distorted by the overflow of the layers. For this reason, only the contour was evaluated.

In the case of such a small sample, improper adjustment of the print parameters may cause the layer diameter to fluctuate. The result is also affected by the overall dynamics of the printer and the rheological properties of the used materials.

### 3.5. Experimental Determination of Material Density

An attempt was also made to change the density of the material. A profile line was drawn across the sample, on which the change in the gray value was evaluated. Figure 7 shows the sample and the profile line for assessing the difference in the gray value and the change in the gray value (bottom). Only the area of the profile line passing through the sample material (gray bounded area) was evaluated.

Table 6 displays that the non-degraded sample showed a higher gray value than solid and porous degraded samples. Therefore, the percentage change of degraded solid samples to non-degraded solid samples was evaluated along with the percentage change from degraded porous to non-degraded porous samples. After recalculating the percentage change in values, the porous degraded samples were shown to have a more remarkable percentage change in values than solid ones.

### 3.6. Evaluation of Mechanical Testing

Figure 8 shows the force dependence on the crosshead position for non-degraded material, SS. As part of the mechanical tests, the tests on solid and porous non-degraded samples were completed after pressing to 1.5 mm or reaching the maximum set value of force.

Figure 9 shows the dependence of force on the position of material degraded in SS.

The results show that for degraded samples, the average stress for individual groups at the end of the mechanical test (max. force, compression) are higher or equal to the pressure for non-degraded samples. The most significant difference is in the PBS solution, where the percentage change for porous samples reached 44% and for solid samples, it was 27% (see Table 7).

To obtain Young´s modulus of elasticity (E), the forces were converted to stress (MPa), and the crosshead displacement to relative compression was achieved. Finally, the following formula was used to determine the relative compression:E = (∆l_0_/l_0_) × 100,(3)
where in our application, l_0_ is the original height of the sample (before deformation), and ∆l_0_ represents the change in height. The stress calculation (σ) is generally defined as the force (F) acting on the surface (A):σ = F/A [MPa](4)

In addition to the force values obtained from the software measurements, the sample areas were also necessary to calculate the stress. Therefore, data obtained by analysing samples using industrial CT were used to determine individual areas. First, it was determined using the percentage ratio between the volume of solid and porous samples. Subsequently, their areas were calculated (Table 8).

By performing these calculations, it was possible to plot the trajectory of individual samples subjected to mechanical compressive stress. The following graph (Figure 10) shows the mechanical tests of compressive strength depending on the mentioned stress and relative compression. Each chart plots 15 samples of a given type (solid or porous) and shows the name of the medium in which the respective samples were degraded. At the end of the graph are the trajectories of the non-degraded samples.

Individual E values of a given group (e.g., SS porous, non-degraded solid samples, etc.) were tested for outliers using the Grubbs test. One sample degraded in Hanks’ solution and one in saline solution were excluded from test analysis.

In the attached Table 9, together with the average values of E, standard deviations are displayed, which were found by calculations.

Table 9 shows that the results of all degraded samples show a lower modulus of elasticity. The most significant change occurred in the case of samples that degraded in PBS solution, and the smallest was in the examined samples which were placed in Hanks’ solution. If we consider the standard deviations, then some values of the solid samples overlap. However, this does not apply to porous samples.

Looking at Figure 9, it is difficult to determine the area that would indicate the transition between the elastic and plastic areas of the material, which we subjected to the compressive strength test. Trajectories of such a form indicate the fragility of the material, which also results from the composition of our mixture. Of course, the influence of the solutions, the temperature, and the absorption of the solutions by the samples in which they degraded must also be considered [15].

## 4. Discussion

Comparing the study outcomes with the published results is strenuous because the material used is a novel and unique material developed at the TUKE workplace. However, it is possible to compare at least partial results.

The use of a combination of polymer and ceramic material could promote the utility of this material for bone regeneration applications.

Micro- and nano-HA is added to polymers to form not only materials with mechanical properties comparable to those of bone, but also that act as a support for bone formation. Furthermore, the addition of bioactive phases to bioresorbable polymers can change the degradation behavior of composite scaffolds [16].

The polymeric material embedded in the culture media affected the pH of these solutions. In principle, it can be assessed that the material changed the pH of the solutions to acidic values. This was most visible when the material was immersed in saline solution. PBS solution showed the most stable environment. The effect of pH on the material and vice versa can affect the rate of degradation of the material. In a study by Lebo et al. [6], they monitored PLA material degradation in PBS at 37 °C. Thus, the pH value has a great influence on the degradation rates.

Our analyses evaluated the weight loss of samples due to degradation of material in different solutions. The weight loss was calculated in five samples randomly selected after degradation from their respective solution and compared to the median obtained from the entire sample set for that solution. Weight loss was demonstrated only in samples from PBS solution. Most studies evaluate weight loss for individual composite components (PLA, PHB). Zhuikov et al., rated weight loss from PLA, PHB, and a 50:50 blend (PLA/PHB) in PBS solution at a pH of 7.4 and at 37 °C. Graphical results showed that the PLA/PHB blend had a weight loss of 10%. The low weight loss for the composite proposed by us may be due to the fact that the ceramic component has three times the density compared to polymers and the component that degrades is the PLA [5].

Mechanical tests show that the average tension for full degraded samples is 185–235 MPa and for porous degraded samples it is 180–206 MPa, depending on the solution used. For non-degraded full samples it was 185 MPa and for non-degraded porous samples it was 143 MPa. Niaza et al. performed mechanical tests on a PLA:HA composite (wt. ratio 85:15), with the Young’s modulus of elasticity found to be 2.8 ± 0.2 GPa with an average maximum tension of 50 MPa [14].

Abeykoon et al. performed mechanical pressure tests on samples made of PLA and CFR-PLA (CFR: carbon fibre reinforced) with 100% infill. The reasonable tension for PLA reached 973 MPa and for CFR-PLA it was 1254 MPa. The compressive modulus for PLA was 1261 MPa and for CFR-PLA it was 1290 MPa. Compressive stress for PLA was 42 MPa and for CFR-PLA it was 332 MPa [17].

Vukasovic et al., tested samples made from PLA under pressure, and the graphical results show that the maximum tension under pressure reached 70 MPa and the Young’s modulus of elasticity reached a maximum at a pressure of 1600 MPa [18].

Hsueh et al. performed mechanical pressure tests on samples made from PLA, with the Young’s modulus of elasticity reaching 3–6 GPa [19].

Meyva-Zeybek et al. conducted mechanical pressurized tests on samples made from PLA, with a maximum compressive tension reaching 65 MPa and the Young’s modulus of elasticity reaching approximately 3.2 GPa [20].

The composite we have developed and presented is composed of PLA:PHB:HA:TCP; therefore, the results obtained by us are not comparable to the results published so far.

## 5. Conclusions

Many biomaterials are currently being tested. In addition, other new types are constantly being developed, which must be subjected to various analyses to discover and eliminate side effects that could cause fatal consequences in an individual’s body.

Thanks to several analyses of the experiments so far, the following results have been reached. The biodegradation process of the PLA/PHB/HA/TCP material mixture is slow. The analysis showed that the increase in weight, and thus the absorption of the solution by the samples, is higher in the case of porous structures. The speed of the degradation process is closely related to this because it has been shown that a more dynamic process takes place in the mentioned porous samples, which was confirmed in the CT analysis by more notable changes in material density and calculations of weight loss. If implanted in the bones, HA and TCP as natural components of the bone could promote osteoconductivity or new bone formation. The PLA and PHB components would be degraded, while HA and TCP would support new bone formation.

The selected material is also suitable for further investigations because a static compressive strength test showed us that no sample cracked; there was only a gradual compression. However, while this may speak of its good mechanical strength properties, which are extremely important in implantation, we cannot generalize these properties, because they need to be supplemented by further tests, e.g., mechanical tensile testing and other biological tests.

## Figures and Tables

**Figure 1 polymers-14-05441-f001:**
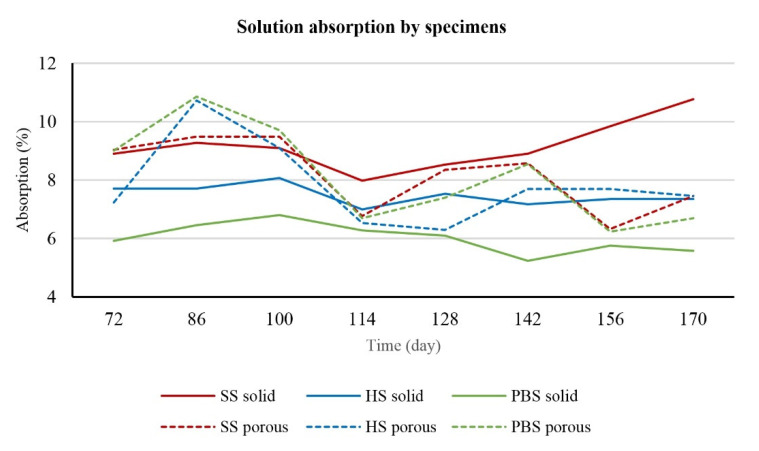
Comparison of solution absorption of different specimens.

**Figure 2 polymers-14-05441-f002:**
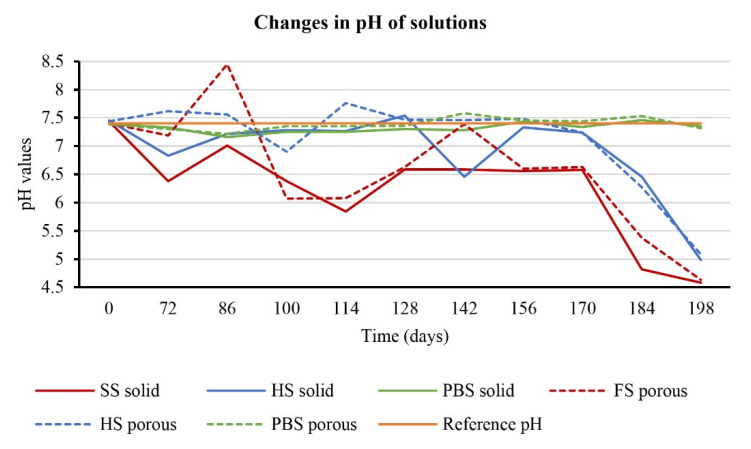
The trajectory of changes in the pH values of individual solutions for 198 days.

**Figure 3 polymers-14-05441-f003:**
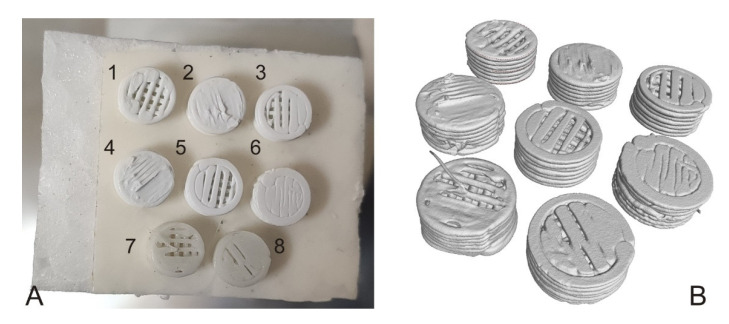
Samples for CT scanning; (**A**) sample storage (1—HS por; 2—HS solid; 3—PBS por; 4—PBS solid; 5—SS por; 6—SS solid; 7—non-degraded porous sample; 8—non-degraded solid sample); (**B**) visualisation of surface reconstruction of individual samples.

**Figure 4 polymers-14-05441-f004:**
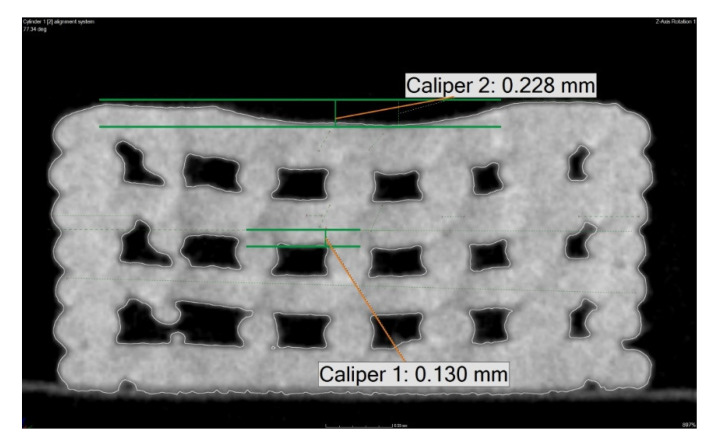
Layer drop observed in degraded samples.

**Figure 5 polymers-14-05441-f005:**
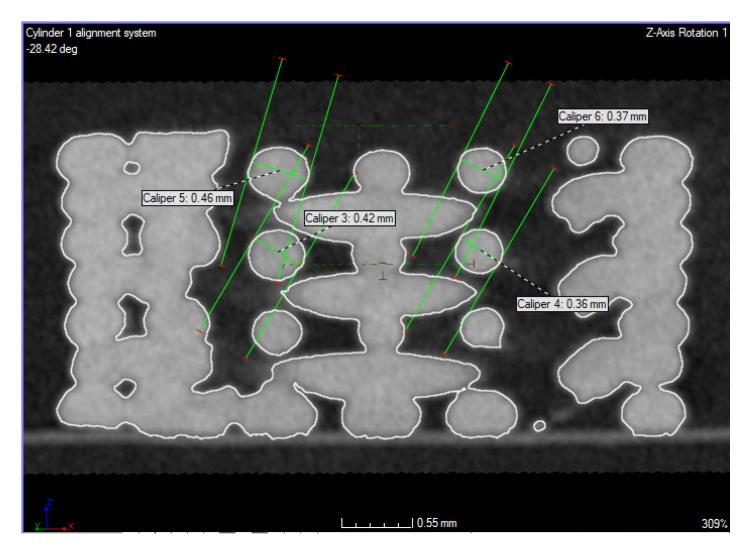
Inner layer diameter range and outer layer diameter range.

**Figure 6 polymers-14-05441-f006:**
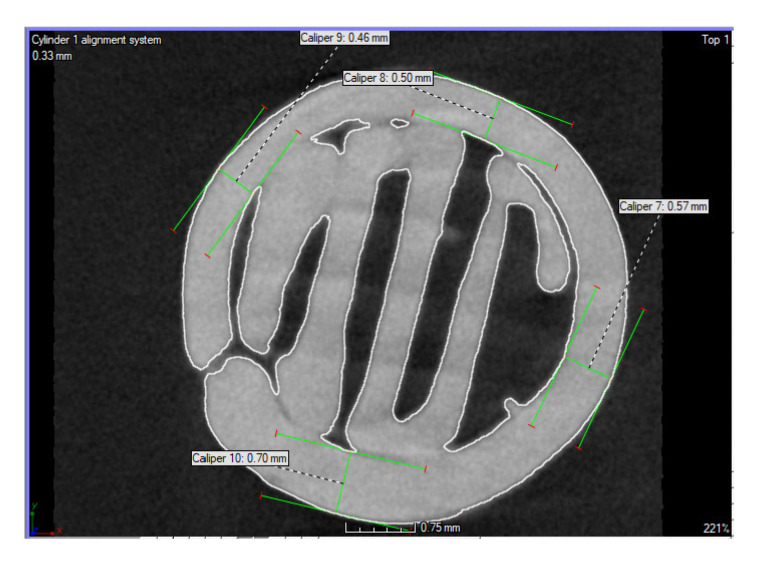
Inner layer diameter range and outer layer diameter range.

**Figure 7 polymers-14-05441-f007:**
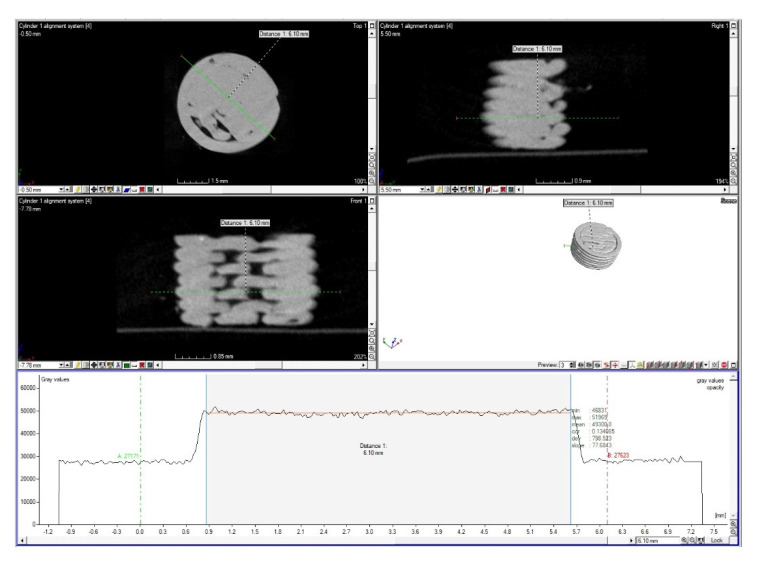
Values of gray value along the profile line.

**Figure 8 polymers-14-05441-f008:**
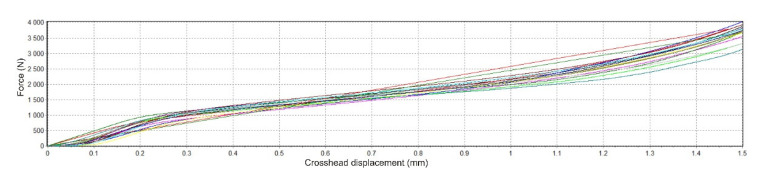
The trajectory of mechanical pressure testing for non-degraded samples in SS.

**Figure 9 polymers-14-05441-f009:**
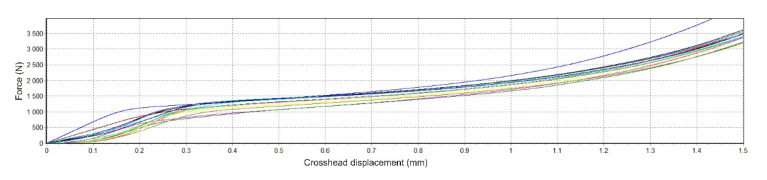
The trajectory of mechanical testing of degraded samples in SS.

**Figure 10 polymers-14-05441-f010:**
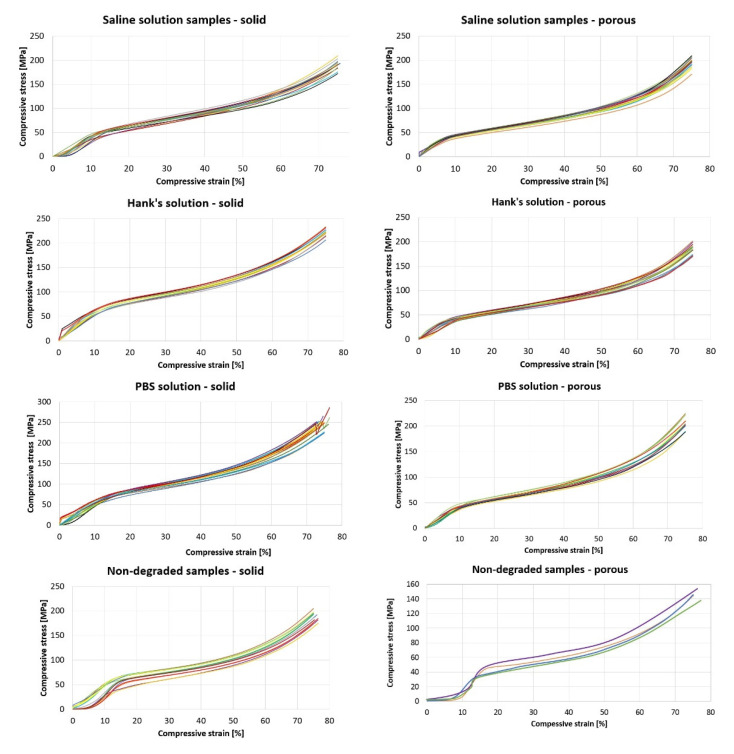
Trajectories for the mechanical pressure test for all samples.

**Table 1 polymers-14-05441-t001:** Thermo-Physical Properties of Materials.

	Density [g/cm^3^]	Melting Point [°C]	Melt Flow Index [g/10 min.]	Molecular Weight [g/mol]
**PLA**	1.24	170–180	10	57.06
**PHB**	1.19	180–190	3	86.09
**HA**	3.14	1650	n.a.	502.3
**TCP**	3.076	1670	n.a.	310.18

**Table 2 polymers-14-05441-t002:** Thermo-Physical Properties of Material PLA/PHB.

	Glass Transition Temperature [°C]	Crystallization Temperature [°C]	Melting Point [°C]	Crystallinity Degree [%]	Density [g/cm^3^]
**PLA/PHB**	61.05	96.5	167.4	22.5	1.23

**Table 3 polymers-14-05441-t003:** Percentage increase in weight of individual groups of samples.

	Solid Samples (%)	Porous Samples (%)
Saline solution	8.24	8.50
Hanks’ solution	7.15	7.43
PBS	6.08	8.19

**Table 4 polymers-14-05441-t004:** Weight loss of material in individual solutions.

	Saline Solution		Hanks’ Solution		PBS	
Weight loss [%]	solid	porous	solid	porous	solid	porous
	−0.82	−0.45	−0.35	−0.11	0.52	1.74

**Table 5 polymers-14-05441-t005:** Data on individual samples obtained from metroromography.

	Volume (mm^3^)	Surface (mm^2^)	Sample Radius (mm)
SS solid	41.42	158.06	2.51
SS porous	32.82	181.39	2.46
HS solid	43.22	145.60	2.46
HS porous	31.82	180.47	2.44
PBS solid	45.50	136.28	2.50
PBS porous	31.45	178.95	2.45
Non-degraded solid	37.81	133.91	2.44
Non-degraded porous	29.88	162.14	2.40

**Table 6 polymers-14-05441-t006:** The change in density of gray.

	Gray Value	Percentage Change (%)
SS solid	48,399	1.83
SS porous	46,870	4.39
HS solid	48,871	0.87
HS porous	48,086	1.91
PBS solid	49,220	0.16
PBS porous	47,600	2.90
Non-degraded porous	49,300	-
Non-degraded solid	49,022	-

**Table 7 polymers-14-05441-t007:** Mechanical pressure test values.

	Stress (MPa)	Percentage Change (%)
SS porous	183.3316	28.3671
HS porous	180.584	26.4433
PBS porous	206.0892	44.3018
Non-degraded porous	142.8182	0
SS solid	185.424	0.101216
HS solid	198.1593	6.76003
PBS solid	235.7762	27.0264
Non-degraded solid	185.6119	0

**Table 8 polymers-14-05441-t008:** Sample area values.

	Surface of Solid Sample (mm^2^)	Surface of Porous Sample (mm^2^)
Non-degraded	18.70	14.78
Saline solution	19.79	15.20
Hanks’ solution	19.01	13.99
PBS solution	19.63	13.57

**Table 9 polymers-14-05441-t009:** Modulus of elasticity of individual tapes of samples.

Modulus of Elasticity in Compression (MPa)				
	Saline Solution	Hanks’ Solution	PBS	Non-Degraded
Solid samples	508.82 ± 11.62	524.53 ± 13.4	475.41 ± 11.57	536.21 ± 22.69
Porous samples	415.01 ± 8.83	427.80 ± 8.33	354.12 ± 22.19	452.83 ± 14.45

## Data Availability

Not applicable.

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
