# Peer review of "Impact of In Vitro Degradation on the Properties of Samples Produced by Additive Production from PLA/PHB-Based Material and Ceramics"

_polymers, 2022, doi:10.3390/polym14245441_

Round 1

Reviewer 1 Report

This work studied the in vitro degradation behavior of PLA/PHB/ceramics material. The overall structure of the article lacks rationality and there are many grammatical errors. The results section only shows and describes the experimental results, and the author does not analyze and discuss each result from the perspective of chemical reactions (degradation), which I think is necessary. In addition, many sample numbers in the text are confused and difficult to understand.

There are some notes:

1.       Line 13 “…a mixture of polymers of polymers of polylactic acid (PLA)”. Check this sentence.

2.       The pilot biological tests were carried out, but it is not mentioned in the Abstract.

3.       Line 17 “PBS”. The abbreviation should be given its full name the first time it appears.

4.       There are many contents in the Introduction part that are not closely related to the research content in this work, so it is suggested to simplify. Please supplement the need to study these performance parameters in this work.

5.       The detailed parameters of PLA/PHB material should be provided, such as melting temperature and density etc. which is important to determination the parameters of FDM-base additive technology.

6.       Line 110 “…in a ratio of 85:15”. Is this ratio mass ratio?

7.       Line125 “The dimensions of the samples were defined based on several analyzed studies”. Is the study referenced here the author’s previous work or other’s literature? References should be given.

8.        Line 126. “sued” should be revised as “used”.

9.       Line 142 The shock treatment with a stirring speed of 18 rpm is given to mimic which physiological conditions in the human body?

10.    How was the 37 °C temperature maintained, and how to ensure the three solutions not volatilize during the 170 days of monitoring so as not to affect the experimental results. After all, the solvent doesn't change much according the author’s result. In addition, in the test item is the loss of solvent considered during the sampling and weighing process?

11.    Line 151 “… the pH was adjusted and adjusted in all solutions using a Metter Toledo pH meter”. Check this sentence.

12.    The parameters in Table 1 have been given in the experiment section, and there is no need to list them separately.

13.    Line 207, “nad” should be revised as “and”.

14.    Line 233. Is the word “aby” correctly used?

15.    Why is the weight loss of some samples negative?

16.    Line 241 “Influence of material on pH”, whose pH?

17.    Figure 3 should show a comparison of each sample before and after degradation, not just one, because each printed sample surface itself is different. The reader cannot see a comparison before and after degradation, after all, this is a very small change from the picture. Also check the name of sample # 8, is “8-non-degraded sample plan” or “8-non-degraded sample solid”?

18.    Line 262 what is HR and FR?

19.    Figure 8 and Figure 9 is not clear.

20.    What are the “NEDEGR. solid” and “NEDEGR. porous” in Table 4 as well as the “NON-DEG. solid” and “NON-DEG. porous” in Table 5, but “Non-degraded porous” and “Non-degraded solid” in Table 6. They should be defined and unified. Moreover, whether the text in the table is bold or not should be consistent in different Tables.

21.    Line 404, what is CMSCs?

22.    The Evaluation of pilot biological tests is too ordinary. If this performance of the PLA/PHB composite has been reported, this test is not necessary. In addition, the author does not explain what kind of sample is used for testing, the raw PLA/PHB material is granular, how is the test carried out? More importantly, the authors used the PLA/PHB that had not been degraded, so that content is not covered in the title.

23.    The Discussion section should be an analysis and discussion of the author's experimental results rather than a restatement of the background.

24.    It is not needed to repeat the idea and experiment of the work in the first and second paragraphs of the conclusion.

25.    The test method and parameters of mechanical properties are not given in this paper.

Author Response

REVIEWER 1

Comments and Suggestions for Authors

This work studied the in vitro degradation behavior of PLA/PHB/ceramics material. The overall structure of the article lacks rationality and there are many grammatical errors. The results section only shows and describes the experimental results, and the author does not analyze and discuss each result from the perspective of chemical reactions (degradation), which I think is necessary. In addition, many sample numbers in the text are confused and difficult to understand.

There are some notes:

  1. Line 13 “…a mixture of polymers of polymers of polylactic acid (PLA)”. Check this sentence.

The sentence was corrected to: a mixture of polymers of polylactic acid (PLA)

  1. The pilot biological tests were carried out, but it is not mentioned in the Abstract.

Chapter 3.7 Evaluation of pilot biological tests was removed

  1. Line 17 “PBS”. The abbreviation should be given its full name the first time it appears.

The abbreviation was added in text: Phosphate-buffered saline

  1. There are many contents in the Introduction part that are not closely related to the research content in this work, so it is suggested to simplify. Please supplement the need to study these performance parameters in this work.

The Introduction was rewritten

  1. The detailed parameters of PLA/PHB material should be provided, such as melting temperature and density etc. which is important to determination the parameters of FDM-base additive technology.

The next text was added:

Detailed parameters about PLA/PHB material have been added. We agree that the melting temperature of the investigated material has a significant impact on the 3D printing process. However, as stated by several studies in the given research area (Hwang et al., Benwood et al., Hsueh et al.), the temperature of the extruder/nozzle is significantly higher than the melting temperature of the given material. This is due to the smaller nozzle diameter (typically 0.4 mm nozzle diameter) used in these types of 3D printers. For this reason, during 3D printing, the temperature is higher than the melting temperature of the material under investigation. This fact is also confirmed by the melting temperature of pure PLA material (melting temperature from 170 to 180), but during 3D printing the material heats up to a temperature of around 210 (values given by filament manufacturers and scientific articles).

References

Hwang S, Reyes EI, Moon KS, Rumpf RC, Kim NS. Thermo-mechanical characterization of metal/polymer composite filaments and printing parameter study for fused deposition modeling in the 3D printing process. Journal of Electronic Materials. 2015 Mar;44(3):771-7.

Benwood C, Anstey A, Andrzejewski J, Misra M, Mohanty AK. Improving the impact strength and heat resistance of 3D printed models: structure, property, and processing correlationships during fused deposition modeling (FDM) of poly (lactic acid). Acs Omega. 2018 Apr 23;3(4):4400-11.

Hsueh MH, Lai CJ, Liu KY, Chung CF, Wang SH, Pan CY, Huang WC, Hsieh CH, Zeng YS. Effects of Printing Temperature and Filling Percentage on the Mechanical Behavior of Fused Deposition Molding Technology Components for 3D Printing. Polymers. 2021 Aug 29;13(17):2910.

  1. Line 110 “…in a ratio of 85:15”. Is this ratio mass ratio?

- Yes, "85:15" represents the weight ratio of PLA and PHB in the composite material (85 weight ratio for PLA and 15 weight ratio for PHB)

  1. Line125 “The dimensions of the samples were defined based on several analyzed studies”. Is the study referenced here the author’s previous work or other’s literature? References should be given.

The reference was added

  1. Line 126. “sued” should be revised as “used”.

The word was corrected

  1. Line 142 The shock treatment with a stirring speed of 18 rpm is given to mimic which physiological conditions in the human body?

Thanks for warning. It should not be 18 but 150RPM and we based it on the study:

Zhuikov, V.A.; Akoulina, E.A.; Chesnokova, D. V.; Wenhao, Y.; Makhina, T.K.; Demyanova, I. V.; Zhuikova, Y. V.; Voinova, V. V.; Belishev, N. V.; Surmenev, R.A.; et al. The Growth of 3T3 Fibroblasts on PHB, PLA and PHB/PLA Blend Films at Different Stages of Their Biodegradation In Vitro. Polymers 2021, 13, 1–23, doi:10.3390/POLYM13010108.

  1. How was the 37 °C temperature maintained, and how to ensure the three solutions not volatilize during the 170 days of monitoring so as not to affect the experimental results. After all, the solvent doesn't change much according the author’s result. In addition, in the test item is the loss of solvent considered during the sampling and weighing process?

The ESCO CCL-170B-8 device has a thermal fluctuation of 0.2°C from the set temperature value. The temperature is controlled and controlled by the device. During the experiment, the beakers were closed and there was only minimal evaporation of the solution. It was not necessary to top up the solution during the entire experiment. When weighing the samples, the pH of the solution was measured at the same time, and in case of a change in pH, the pH of the solution was corrected. The Binder device from Otto Bock was only used for drying the samples.

  1. Line 151 “… the pH was adjusted and adjusted in all solutions using a Metter Toledo pH meter”. Check this sentence.

The sentence was improved

  1. The parameters in Table 1 have been given in the experiment section, and there is no need to list them separately.

The Table 1 was removed from the text, the missing parameters are mentioned in the text

  1. Line 207, “nad” should be revised as “and”.

The word was corrected

  1. Line 233. Is the word “aby” correctly used?

The word was corrected

  1. Why is the weight loss of some samples negative?

Weight loss was calculated from randomly selected 5 samples after degradation from the respective solution and compared to the median obtained from the entire set of samples for that solution. Due to the fact that it is a selection of samples from the total set, the value of the median weight for the selected sample after degradation is greater than the value of the median for the entire set of samples before degradation. Due to further tests, it was not possible to use the entire set of samples after degradation.

  1. Line 241 “Influence of material on pH”, whose pH?

The sentence was improved

  1. Figure 3 should show a comparison of each sample before and after degradation, not just one, because each printed sample surface itself is different. The reader cannot see a comparison before and after degradation, after all, this is a very small change from the picture. Also check the name of sample # 8, is “8-non-degraded sample plan” or “8-non-degraded sample solid”?

The text was improved:

CT analysis was performed using a Metrotom 1500 from Carl Zeiss (Germany) after the experiment. Scanning was performed at a resolution of 24.3 µm. From each material and type of sample (full, porous), 1pc was scanned and 1 non-degraded sample was taken as a reference. Figure 3. shows the positioning of samples during scanning and surface reconstruction of all samples.

The figure caption was improved

  1. Line 262 what is HR and FR?

The text was improved

  1. Figure 8 and Figure 9 is not clear.

The quality of figures was improved

  1. What are the “NEDEGR. solid” and “NEDEGR. porous” in Table 4 as well as the “NON-DEG. solid” and “NON-DEG. porous” in Table 5, but “Non-degraded porous” and “Non-degraded solid” in Table 6. They should be defined and unified. Moreover, whether the text in the table is bold or not should be consistent in different Tables.

The text was improved

  1. Line 404, what is CMSCs?

The text was removed

  1. The Evaluation of pilot biological tests is too ordinary. If this performance of the PLA/PHB composite has been reported, this test is not necessary. In addition, the author does not explain what kind of sample is used for testing, the raw PLA/PHB material is granular, how is the test carried out? More importantly, the authors used the PLA/PHB that had not been degraded, so that content is not covered in the title.

 The Evaluation of pilot biological tests was removed

  1. The Discussion section should be an analysis and discussion of the author's experimental results rather than a restatement of the background.

The discussion was improved

  1. It is not needed to repeat the idea and experiment of the work in the first and second paragraphs of the conclusion.

The text was improved

  1. The test method and parameters of mechanical properties are not given in this paper.

The test methods and parameter are added to text

Reviewer 2 Report

1.In Figure 1. Comparison of solution absorption aby different specimens correct typo aby with by.Similary why there is retention of absorption of water after 86 days?comments on physicochemical properties contributing for it?

2.Improve quality of images such as Fig.7,8,9,10?   Major revision and rewriting introduction and results easy to follow to reader.

Author Response

REVIEWER 2

Comments and Suggestions for Authors

  1. In Figure 1. Comparison of solution absorption aby different specimens correct typo aby with by.Similary why there is retention of absorption of water after 86 days? comments on physicochemical properties contributing for it?

The Figure. 1 camption was improved

Thank you for the notice. We will investigate this increase in absorption in the next study. Since this is a new composite material and it is currently not possible to verify our results with relevant studies.

  1. Improve quality of images such as Fig.7,8,9,10?   Major revision and rewriting introduction and results easy to follow to reader.

The figure quality was improved, the template of journal limit the figure size. The introduction and the discussion were rewrited.

Round 2

Reviewer 2 Report

I accept it.